# Learning to Edit Visual Programs with Self-Supervision

**R. Kenny Jones**
Brown University
russell_jones@brown.edu

**Renhao Zhang**
University of Massachusetts, Amherst
renhaozhang@cs.umass.edu

**Aditya Ganeshan**
Brown University
aditya_ganeshan@brown.edu

**Daniel Ritchie**
Brown University
daniel_ritchie@brown.edu

## Abstract

We design a system that learns how to edit visual programs. Our edit network consumes a complete input program and a visual target. From this input, we task our network with predicting a local edit operation that could be applied to the input program to improve its similarity to the target. In order to apply this scheme for domains that lack program annotations, we develop a self-supervised learning approach that integrates this edit network into a bootstrapped finetuning loop along with a network that predicts entire programs in *one-shot*. Our joint finetuning scheme, when coupled with an inference procedure that initializes a population from the *one-shot* model and evolves members of this population with the edit network, helps to infer more accurate visual programs. Over multiple domains, we experimentally compare our method against the alternative of using only the *one-shot* model, and find that even under equal search-time budgets, our editing-based paradigm provides significant advantages.

## 1 Introduction

People seldom write code with a linear workflow. The process of authoring code often involves substantial trial-and-error: possibly correct programs are evaluated through execution to see if they raise exceptions or break input-output assumptions. When an error is identified, an edit is made, and this process is repeated. It is difficult to imagine writing any moderately complex program in a *one-shot* paradigm, without being able to debug intermediate program versions.

The field of program synthesis studies how to automatically infer a program that meets an input specification [13]. In this work, we consider the sub-problem of visual program induction (VPI), where the input specification is a visual target (e.g. an image) and the goal is to find a program whose execution recreates the target [33]. This task has numerous applications across visual computing disciplines, including reverse-engineering, structure analysis, manipulation, and generative modeling.

This problem area has garnered significant interest, with many works exploring *learning*-based solutions. For domains with annotated data, supervised approaches perform well [41, 43]. For domains that lack program annotations, a variety of unsupervised and self-supervised learning paradigms have been proposed [19, 34, 38, 45]. Moreover, initial investigations have explored the capabilities of Large Language Models for solving simple VPI tasks [3].

Though these prior approaches have made impressive progress, a common limitation is that they operate within the aforementioned *one-shot* paradigm. For instance, when using an autoregressive network these systems will condition on a visual target and iteratively predict program tokens until

38th Conference on Neural Information Processing Systems (NeurIPS 2024).

completion. While this sequential inference procedure is sometimes wrapped in a more complex outer-search (e.g. beam-search or sequential Monte Carlo [10]), is allowed to reason over partial program executions [25], or is given access to executor-gradients that guide an inner-loop search [12, 45], all of these paradigms are distinct from how *people* typically write programs.

In this work, we present a model that learns how to edit visual programs in a goal-directed manner. Our network consumes a complete input program, this program's executed state, and a visual target. It then proposes a local edit operation that modifies the input program to better match the target. In contrast with *one-shot* approaches, this framing allows our network to explicitly reason over a complete program and its execution, in order to decide how this program should be modified.

We train our network without access to any ground-truth program annotations. To accomplish this, we propose an integration of our edit network with prior self-supervised bootstrapping approaches for *one-shot* VPI models [19]. During iterative finetuning rounds, we source paired training data for our edit network by first constructing pairs of start and end programs, and then using a domain-aware algorithm to find a set of edit operations that would bring about this transformation. This process jointly finetunes both our edit network and a *one-shot* network, and we propose an integrated inference algorithm that leverages the strengths of both of these paradigms: the *one-shot* model produces rough estimates that are refined with the edit network. We find that this joint self-supervised learning set-up forms a virtuous cycle: the *one-shot* model provides a good initialization state for the edit network, and the edit network improves inner-loop inference, creating better bootstrapped training data for the *one-shot* model.

We experimentally compare the effectiveness of integrating our edit network into this joint paradigm against using *one-shot* models alone. Controlling for equal inference time, over multiple visual programming domains, we find that using the edit network improves reconstruction performance. Moreover, we find that the reconstruction gap between these two paradigms widens as more time is spent on test-time program search. Further, we demonstrate our method performs remarkably well even with very limited data, as learning how to edit is an inherently more local task compared with learning how to author a complete program. Finally, we run an ablation study to understand and justify our system design.

In summary, we make the following contributions: (1) A model that learns how to predict local edits that improve visual programs towards a target. (2) A self-supervised learning paradigm that jointly trains an edit network and a *one-shot* network through bootstrapped finetuning.

We release code for our experiments at: https://github.com/rkjones4/VPI-Edit

## 2   Related Work

**Visual Program Induction** There has been growing interest in works that aim to infer structured procedural representations that explain visual data [33]. While some research has investigated geometric heuristics to search for a well-reconstructing program [9, 42], most methods employ learned networks to guide this search. For visual programming domains that come with annotations, networks can be trained with ground-truth program supervision [1, 17, 41, 43].

However, most visual domains of interest lack such annotated data. As a result, a host of techniques have been investigated for this unsupervised setting, including: reinforcement learning [10, 34], differentiable execution architectures [20, 31, 36, 44, 45], learned proxy executors [8, 38], or boot-strapping methods [11, 15, 19]. All of these works operate within the aforementioned *one-shot* paradigm. Of note, SIRI investigates how analytical code rewriting operations can improve VPI networks in a bootstrapped learning paradigm [12]. Our system shares a similar motivation in that we aim to rewrite visual programs in a goal-directed fashion. However, instead of modifying programs with domain-specific fixed operations (e.g. differentiable parameter optimization), we explore a generalized alternative by introducing a network that learns how to edit programs.

Our edit network reasons over the visual execution produced by an input program to decide how the program should be edited. The idea of reasoning over program executions to improve search has been successfully demonstrated for both general program synthesis problems [6, 47] and visual program induction [10, 25]. However, different from our approach, which predicts a local edit that modifies a complete program, these approaches reason over the executions of partial programs in order to better guide auto-regressive synthesis (i.e. they largely operate in a *one-shot* paradigm).

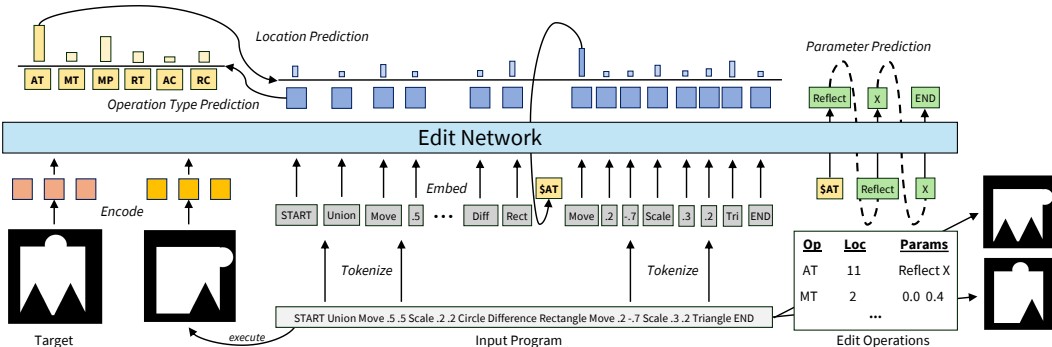

Figure 1: We design a network that learns how to locally edit an input program towards a target. It first predicts what type of edit operation should be applied, then it predicts where that edit operation should be applied, and finally it autoregressively samples any parameters the edit operation requires.

**Program Repair** A number of program synthesis methods have been proposed that learn how to repair or fix programs for domains where ground-truth programs are available. SED interleaves a series of synthesis, execution and debugging steps in order to improve synthesis of Karel programs from input/output examples [14]. Related approaches have explored learning how to 'fix' programs end-to-end by manipulating latent-space encodings of programs under a fixed decoder for the RobustFill domain [2]. While our method shares a similar motivation with these works, we demonstrate the efficacy of our approach on more complex visual programming domains, we don't assume access to ground-truth program annotations, and our edit network only predicts a single local edit operation at each step, instead of rewriting an entire program.

With the growing attention around the abilities of Large Language Models (LLMs), a number of recent works have explored how LLMs can be used to fix programs from input/output examples [5, 23, 24, 35]. Though differing in details, the typical formulation these methods take involves presenting an LLM with a previous program version, and asking it to either (i) debug exceptions or (ii) modify program behavior in light of input/output mismatches. While these initial forays show promise, LLMs have not yet been able to write complex visual programs (in part due to poor visual reasoning capabilities), and even for more general program synthesis tasks the performance gains of code-editing LLMs are not definitive [26].

## 3 Method

In this section, we present our approach for learning how to edit visual programs. First we formalize our task of unsupervised visual program induction. For a particular domain, we are given a domain-specific language (DSL) $L$ and an executor $E$ that converts programs $z$ from $L$ into visual outputs $x$. Given visual inputs from a target visual dataset that lacks program annotations, $x^* \in X^*$, our goal is to find find $z^* \in L$, such that $E(z^*) \sim x^*$. This measure of similarity is usually checked under a domain specific reconstruction metric $M$.

A general approach employed by prior visual program induction works is to use an autoregressive model (e.g. a Transformer) that is conditioned on a visual encoding to predict a well-reconstructing program: $p(z|x)$. These *one-shot* models iteratively predict the next program token until the program is complete. We present a framework that employs a similar autoregressive model, but instead of predicting a complete program from scratch, we instead predict a local edit that modifies an input program. In the rest of this section, we first present how we design our edit network (Sec. 3.1). Then we discuss our unsupervised training procedure where we jointly finetune an edit network along with a *one-shot* network (Sec. 3.2. Finally, we describe how we combine these networks to search for visual programs (Sec. 3.3).

### 3.1 Edit Network Design

Our edit network $p(e|z, x)$ learns how to predict a local edit operation that improves an input program towards a visual target (see Figure 1). We provide our network with a triplet input state: the tokens of

an input program $z$, this program's executed output $E(z)$, and a visual target $x$. From this state, our network is tasked with predicting an edit operation $e$ that could be applied to the input program.

**Edit Operations.** There are many ways to parameterize the space of possible program edits. We choose to constrain the possible edit operations our network can produce by forcing it to select from a set of local editing operations designed for visual programs. For instance, for functional visual programming DSLs with transformation and combinator functions, we allow for seven different edit operations: modifying a transform's parameters (*MP*), modifying a transform (*MT*), adding a transform (*AT*), removing a transform (*RT*), modifying a combinator (*MC*), removing a combinator (*RC*), or adding a combinator (*AC*). We provide more details in Appendix E. Some of these edit operations do not take in parameters (removing a transform) while others require new parameters (e.g. to modify the parameters of a transform we need to know the new parameters). Each of these edit operations can be applied to a program at a specific token location, and results in a local change. Subsequently, we task our edit network with predicting three items: an edit operation type, a location for that edit operation, and any extra parameters that operation requires.

We design our system with this somewhat constrained edit operation set as it has a number of advantages. First, the application and effect of each edit operation is local; this simplifies the learning task and allows us flexibility at inference time. Moreover, ensuring that edit operations are tied to the semantics of the underlying DSL helps to promote program edits that result in syntactically valid modified programs. We compare our edit operation design against alternative formulations in our experimental results (Sec. 4.5).

**Architecture.** We implement our edit network as a Transformer decoder. This network has full attention over the conditioning information: each visual input (the executed output of the input program and the target) is encoded into a sequence of visual tokens (e.g. with a CNN) and each token of the input program is lifted with an embedding layer.

To predict the edit operation type, we take the output Transformer embedding from the first index of input program sequence. This embedding is sent through a linear layer which predicts a distribution over the possible edit operation types (yellow boxes, Fig. 1).

To predict the edit operation location, we consider the embeddings that the Transformer produces over the tokens of the input program. Each of these location codes is sent through a linear layer, which predicts a value for each operation type. For a chosen operation type, we then normalize these values into a probability distribution across the length of the input program sequence (dark-blue boxes, Fig. 1). This distribution models the likelihood of where a specific edit operation type should be applied.

Finally, we use our network to autoregressively sample any extra parameters that a chosen edit operation might require. To accomplish this, we first slightly reformat the input program by inserting a special 'sentinel token' [29] associated with the chosen edit operation in two places: (1) at the specified edit operation location and (2) at the end location of the current program (*$AT*, Fig. 1). This 'sentinel' tokens allows the network to know what operation is being applied to which position. Then, starting from the location of the second sentinel token, we can use the network to iteratively generate a sequence of parameter predictions with causal attention-masking, until an 'END' token is chosen (green boxes, Fig. 1).

**Training.** Given an input program, how do we know which edit operations are helpful? If we have access to not only a visual target, but also its corresponding program, we can find a set of edit operations that would transform the input program into this target. We follow this logic to source training data for our edit network: given a start program and an end program, we analytically identify a set of edit operations that would bring about this transformation with a *findEdits* function. We can then convert this set of edit operations into a large set of (input, output) pairs that our network can train on. We provide further details on this algorithm in Appendix E. Once we have sourced paired data, through teacher-forcing we can train our network in a supervised fashion with a cross-entropy loss on the predicted operation type, location, and each parameter token. Though we lack known programs for the target domain of interest, we next discuss a bootstrapped finetuning procedure that provides a work-around for this issue.

**Algorithm 1** Network Training

1: $p(z|x) \leftarrow$ pretrain($L$)
2: $p(e|z,x) \leftarrow$ pretrain($L, p(z|x)$)
3: $P^{\text{BEST}} \leftarrow \{\}$
4: **for** $num\_rounds$ **do**
5: $\quad P^{\text{BEST}} \leftarrow$ Infer($X^*, p(z|x), p(e|z,x)$)
6: $\quad p(z) \leftarrow$ trainGen($P^{\text{BEST}}$)
7: $\quad P^G \leftarrow$ sample($p(z), \{\}$)
8: $\quad P^S \leftarrow$ sample($p(z|x), E(P^G)$)
9: $\quad ES \leftarrow$ findEdits($P^S, P^G$)
10: $\quad p(e|z,x) \leftarrow$ trainEdit($ES$)
11: $\quad p(z|x) \leftarrow$ trainPLAD($P^{\text{BEST}}, P^G$)
12: **end for**

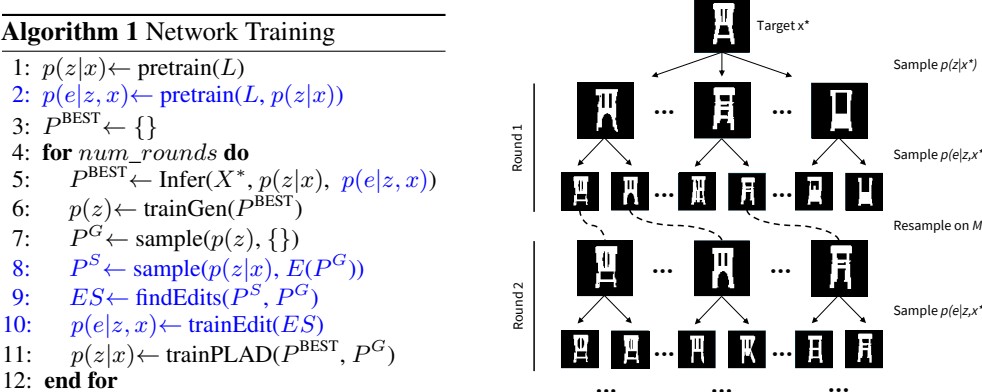

Figure 2: *Left:* our bootstrapping algorithm that finetunes an edit network and a *one-shot* model towards a target dataset. *Right:* our inference algorithm that initializes a population with a *one-shot* model and then mutates it towards a visual target through iterative rounds of edits and resampling.

## 3.2 Learning Paradigm

As we operate in a paradigm where we don't have access to ground-truth programs for our target set $X^*$, we take inspiration from recent self-supervised approaches that employ bootstrapped finetuning for visual program induction [12, 19]. Specifically, we develop an algorithm (Alg. 1) that integrates edit network training into the PLAD finetuning framework.

**PLAD Finetuning.** We begin with an overview of the PLAD method, which is depicted with the black text in Alg. 1 (see [19] for details). At the start of each round, the program inference network $p(z|x)$ is run over the target dataset $X^*$; the results of this inference procedure populate the entries of a best programs data-structure $P^{\text{BEST}}$ according to $M$. Then an unconditional generative model $p(z)$ is trained over the entries of $P^{\text{BEST}}$, and a set of 'dreamed' programs, $P^G$, are sampled from this network. The weights of $p(z|x)$ are then finetuned using paired data sourced from $P^{\text{BEST}}$ and $P^G$. These steps are repeated for a set number of rounds, or until convergence.

**Edit Model Finetuning.** The blue-colored lines in Alg. 1 indicate the modifications we make to the PLAD algorithm to incorporate our edit network. Lines 8-10 explain the training logic. First we use $p(z|x)$ to sample a set of programs $P^S$ conditioned on the executed outputs of the generated programs $P^G$. Treating $P^S$ as the starting points and $P^G$ as the end points, we can then use our *findEdits* operation to find sets of edit operations $ES$ that would realize these transformations. This provides us with paired data that we can use to finetune the weights of the edit network through teacher forcing, as explained in the prior section.

**Synthetic Pretraining.** PLAD finetuning is typically initialized with a synthetic pretraining phase (Alg. 1, line 1). During pretraining, random programs are sampled from $L$, and $p(z|x)$ can be trained on the paired data produced by executing these samples. Similarly, as we discuss in the results section, we find it useful to 'pretrain' the edit network on synthetic data (Alg. 1, line 2). While multiple formulations are possible here, we re-use the same logic shown on lines 8-10, except we replace the set of target programs $P^G$ with random programs sampled from $L$.

## 3.3 Inference Algorithm

With the above procedure we can train our edit network, but how can we use this network to find improved visual programs? This question is not only relevant at test-time, but also impacts bootstrapped training, as we run an inner-loop search to populate the entries of $P^{\text{BEST}}$ (Alg. 1, line 5). As depicted on the right side of Figure 2, we design a search procedure that combines the strengths of the *one-shot* and editing paradigms. This search procedure maintains a population of programs, which are evolved over a number of rounds. The initial population is produced by sampling $p(z|x)$. Then for each round, we use the edit network to sample sets of edits for every program in the current population. We apply each of these sampled edits, and then re-sample the population for the next round according to a ranking based on $M$.

Table 1: Across multiple visual programming domains we evaluate test-set reconstruction accuracy. In all cases, we find that our joint paradigm that integrates an edit network with *one-shot* models outperforms the alternative of using only *one-shot* models.

|  | Layout *cIoU* $\Uparrow$ | 2D CSG *CD* $\Downarrow$ | 3D CSG *IoU* $\Uparrow$ |
|---|---|---|---|
| *OS Only* | 0.94 | 0.156 | 83.3 |
| *OS + Edit (Ours)* | **0.98** | **0.111** | **85.3** |

This formulation has a number of advantages. Instead of starting from a blank canvas, or with random samples, we allow $p(z|x)$ to produce initial rough program estimates. These guesses are then refined through mutations over a series of editing rounds that are all directed at improving similarity towards the visual target. In Section 4.5 we compare this algorithm against alternative formulations. Critically, by applying this joint inference procedure during finetuning we form a virtuous cycle: improving the inference strategy leads to better $P^{\text{BEST}}$ entries, which results in better training data for $p(z|x)$ and $p(e|z,x)$, which in turn allows us to find to better $P^{\text{BEST}}$ entries in subsequent finetuning rounds. Finally, we note that this formulation maintains a nice symmetry between $p(z|x)$ and $p(e|z,x)$: in out joint finetuning algorithm $p(e|z,x)$ trains on sequences sourced from sampling $p(z|x)$, and in this way its training distribution of edit operations well matches the population used to initialize the inference algorithm.

## 4 Results

We evaluate our edit network with experiments over multiple domains. First we describe our experimental design (Sec. 4.1). Then we compare the ability of different methods to accurately infer visual programs in terms of reconstruction performance (Sec. 4.2). We analyze how this performance changes as a function of time spent on inference (Sec. 4.3) or the size of the training target dataset (Sec. 4.4). Finally, we discuss results of an ablation study on our method in Section 4.5.

### 4.1 Experimental Design

We provide a high-level overview of our experimental design. See Appendix D for details.

**Methods.** We compare our approach (*OS+Edit*) against the alternative of using only a *one-shot* model (*OS Only*). As described in Section 3, our approach jointly finetunes an edit network along with a *one-shot* network, and uses both of these networks to infer visual programs (Fig. 2). To control for the added time cost incurred by our inference procedure, we adapt a sampling-based inference loop for the *OS Only* variant, which we find results in a surprisingly strong baseline.

**Domains.** We consider three VPI domains (see App C): Layout, 2D CSG, and 3D CSG. In the Layout domain, scenes are created by placing colored 2D primitives on a canvas, and optionally modifying them by changing their size, location, or forming a symmetry group. In constructive solid geometry (CSG), complex shapes are formed by combining simple shapes with boolean set operations (union, intersection, difference). Our 2D CSG and 3D CSG domains differ in terms of their primitive types (e.g. squares vs cuboids) and the parameterizations of transformation functions: generalizing notions of scaling, translating, rotating, and symmetry grouping from $\mathbb{R}^2$ to $\mathbb{R}^3$.

**Network Details.** For each domain, we implement $p(z|x)$ as a decoder-only Transformer [39] that conditions on a set of visual tokens and predicts up to a maximum sequence length *SL*. Similarly, we implement $p(e|z,x)$ as a Transformer with the same architecture, except that it conditions on (i) two sets of visual tokens and (ii) an input program of length *SL*, and it is only allowed to predict edit parameters up to a length of *EL*. Our visual encoders are all standard CNNs. For Layout we use a 2D CNN that takes in an RGB 64x64 image, for 2D CSG we use a 2D CNN that takes in a binary 64x64 image, and for 3D CSG we use a 3D CNN that takes in a $32^3$ voxel grid.

**Reconstruction Metric.** The reconstruction metric $M$ guides the inference algorithm and also performs early stopping with respect to a validation set. For Layout we use *cIoU*, an intersection over union metric which only counts intersections on color matches [18]. For 2D CSG we use an edge-based Chamfer distance (*CD*) [34]. For 3D CSG we use intersection over union (*IoU*).

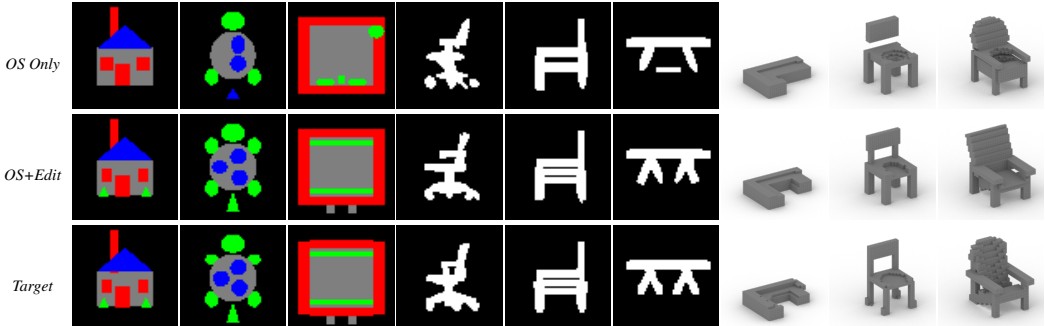

Figure 3: Comparing reconstructions of *one-shot* models (*top*) against our joint approach (*middle*).

**Target Data.** Like prior bootstrapping methods, our finetuning algorithm specializes our networks towards a target dataset of interest, $X^*$, that lacks known programs. For 2D CSG we use shapes from the dataset introduced by CSGNet [34], originally sourced from Trimble 3D warehouse. For 3D CSG we use shapes from the dataset introduced by PLAD [19], originally sourced from ShapeNet [4]. While we use the same test-sets as prior work (3000 / 1000 for 2D CSG / 3D CSG), we find that our method is able to offer good performance with much less training data. In our base experiments, we use 1000/100 train/val shapes for 2D CSG (from 10000 / 3000 available) and and 1000/100 train/val shapes for 3D CSG (from 10000 / 1000 available). For the Layout domain, we use the manually designed scenes sourced from [18] (1000 train / 100 val / 144 test).

## 4.2 Reconstruction Accuracy

We compare our *OS+Edit* approach against *OS Only* on each method's ability to infer visual programs that accurately reconstruct test-set inputs in Table 1. As demonstrated, our joint finetuning paradigm that combines an edit network with a *one-shot* network consistently improves reconstruction performance. In these experiments, we ensure that each method gets to spend the same amount of time on inference by setting search parameters so that the average inference time per shape was equal: $\sim 5$, $\sim 10$, $\sim 60$ seconds per shape for Layout, 2D CSG, and 3D CSG respectively. For *OS Only*, we use a sampling-based inference search where the model samples a population of complete programs for a set number of rounds. Though this approach provides a strong baseline, it was not as effective as combining our edit networks with *one-shot* initializations. In fact, for the 2D CSG domain, our formulation achieves reconstruction scores that surpass the performance of related methods that assume access to executor-gradients. On the 2D CSG test-set, we achieve a Chamfer distance (CD) of 0.111 (lower is better), whereas UCSG-Net [20] gets a CD of 0.320, SIRI [12] gets a CD of 0.260, and ROAP [36] gets a CD of 0.210 . Note that as the DSL, architecture, objective, and inference procedures differ across these various works, it's hard to make any absolute claims from this direct comparison. Nevertheless we would like to emphasize that our method's reconstruction performance on this task is very strong in the context of the related literature. We visualize reconstructions from this experiment in Figure 3, and find that qualitative evidence supports the quantitative trends.

## 4.3 Search Time

While *one-shot* models must author new programs from scratch without execution-feedback, our edit network has the capacity to reason over an input program, compare its execution versus the visual target, and decide how this program should be modified. As such, we hypothesize that integrating our edit network into our inference procedure will be increasingly advantageous over the *OS Only* approach as more time is spent on test-time search. To validate this hypothesis, we explore how the reconstruction gap between these paradigms changes as a function of time spent on search (Figure 4, *left*). For 2D CSG we take a subset of the test-set (300 shapes) and run more rounds of our inference algorithm. As demonstrated, as more time is spent on test-time search (i.e. as the number of rounds increases) the reconstruction gap between *OS Only* and *OS+Edit* grows wider. Moreover, we note that even on the first round there is a gap between the methods, as the *one-shot* network trained in the *OS+Edit* paradigm had access to better $P^{\text{BEST}}$ entries throughout the finetuning process (i.e. the aforementioned virtuous cycle). We present qualitative results that show how the edit network evolves the population of programs towards the visual target in Figure 5.

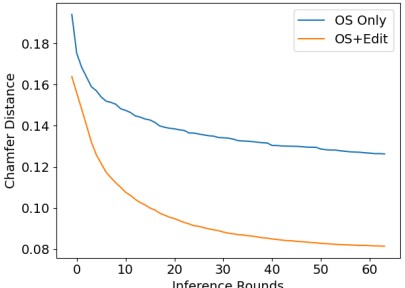 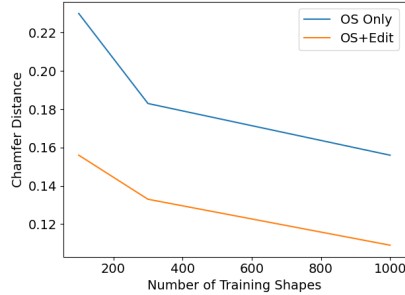

Figure 4: For 2D CSG, we compare reconstruction accuracy (Chamfer distance, lower is better, Y-axis) between using an edit network and using only a *one-shot* network while varying time spent on inference (*left*) and training set size (*right*).

## 4.4 Training with limited data

While both *OS+Edit* and *OS Only* are unsupervised in the sense that they don't have access to any ground-truth program annotations, they do require an input set of visual data to form a target training set. We hypothesize that our edit network will be especially useful for domains with limited data (even limited unannotated data) as the program editing task is inherently more local than trying to author a complete program. Consider for instance that during finetuning, in a *one-shot* paradigm each visual datum can only contribute a single training example, while in our paradigm an entire distribution of edit operations can be sourced by considering the many possible edit paths one could take to transform a start program into an end program. We validate this hypothesize with an experiment where we train versions of these systems while varying the size of the target training set (Fig. 4, *right*). Our joint paradigm offers very strong performance even while finetuning towards an input set of just 100 training shapes, matching the performance of *OS Only* when it has 10x more data.

## 4.5 Method Ablations

We run an ablation experiment to evaluate the design of our system on the Layout domain. We present results of this experiment in Table 2. In the rest of this section we detail all of the alternative formulations we compare against.

**Edit Operations.** Our default edit networks learn how to predict local edit operations from a limited set of options. We compare this paradigm with two alternatives. In the *next program* mode, we task the edit network with predicting all of the tokens of the program that would be created by applying the target edit operation to the input program. In the *final program* mode, we task the edit network with predicting the tokens of the final program associated with the visual target. This formulation was inspired by the success of denoising diffusion models for visual synthesis tasks [16], though in our setting this variant is basically an alternative *one-shot* model with extra conditioning information but with the same target sequences. As demonstrated, neither of these approaches is as performant as our formulation where edits are predicted as local operations. Moreover, predicting an entire program is much slower compared with predicting an edit, so fewer rounds of our inference algorithm can be run with the same search time budget.

**Program Corruption.** We source paired training data for our edit network by constructing (start, end) program pairs and then analytically finding a set of edit operations that would complete this transformation. For an alternative, we can look towards discrete diffusion methods [30, 37, 40, 46]. In our *corruption* variant we take inspiration from these works and design a program corruption algorithm for the Layout domain. This corruption algorithm takes an end program as input, and then samples corruption operations (i.e. inverse edit operations) that can be used as paired data for our edit network (Appendix F). As seen, this alternative formulation was not as performant as our default approach. One reason for this is that it hard to design a corruption process that converts end programs (e.g. $P^G$) into the distribution of programs that we have access to at inference time (e.g. $P^S$). Conversely, by applying our *findEdits* operation on $P^G$ and $P^S$ pairs, we can source paired data for our edit network that *does* match this distribution.

Table 2: Ablation study comparing our method against alternative formulations.

| Method | Final cIoU ⇑ |
|---|---|
| *Ours* | **0.980** |
| *Next program* | 0.941 |
| *Final program* | 0.920 |
| *Corruption* | 0.964 |
| *No FT* | 0.955 |
| *No one-shot FT* | 0.972 |
| *No edit FT* | 0.976 |
| *No edit PT* | 0.953 |
| *Naive OS* | 0.947 |
| *Rand+Edit* | 0.906 |

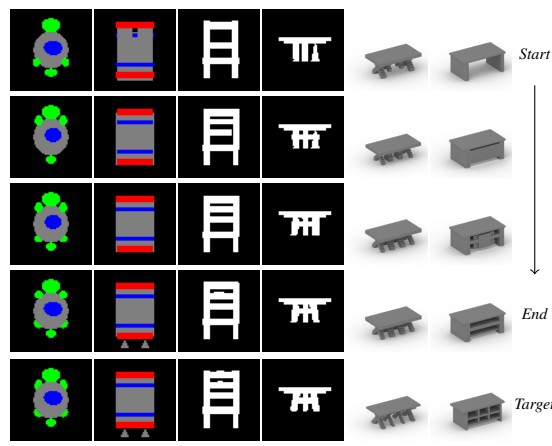

Figure 5: Our inference procedure edits samples from an initial population (*top*) towards a target (*bottom*).

**Pretraining and Finetuning.** In our default version there are three training phases. First, $p(z|x)$ undergoes pretraining on synthetic data. Second, $p(e|z,x)$ undergoes pretraining on synthetic data using samples from $p(z|x)$. Then both of these networks are jointly finetuned with respect to $X^*$. In the *No FT* variant, we don't finetune either network, in *no one-shot FT* we don't finetune $p(z|x)$, in *no edit FT* we don't finetune $p(e|z,x)$, and in *no edit PT* we don't pretrain $p(e|z,x)$. While the performance of our system remains remarkable strong even under these ablations, we get the best results by using all three training phases. Interestingly, for settings where $p(z|x)$ is not specialized for $X^*$, the reconstruction accuracy gap dramatically increases between the best sample in the starting population and the best sample in the final population of our inference procedure. For instance, for the *no one-shot FT* variant, the first round cIoU score is 0.88 which gets increased to 0.972 (0.092 improvement) through the mutations proposed by the edit model, while in our default variant the first round cIoU is 0.925 (an improvement of .055).

**Inference Algorithm.** We compare our inference algorithm with two alternative versions. In *Naive OS* we initialize the first population with $p(z|x)$, and make edits to each population member with $p(e|z,x)$, but we skip the population resampling step according to $M$, and instead apply the highest likelihood edit from $p(e|z,x)$. While the edit network is still helpful in this paradigm (0.022 improvement from the first to the last round), it performs worse compared with our default implementation. In *Rand+Edit*, we remove $p(z|x)$ and instead fill the initial population with random program sampled from $L$. This provides a much worse initialization (0.302 cIoU in the first round), and though our edit network successfully mutates these samples towards the target, better reconstruction performance is gained by combining our edit network with initial guesses from a *one-shot* model.

## 5   Discussion

We have presented a system that learns how to edit visual programs in a goal-directed fashion. We develop a self-supervised bootstrapping approach that allows us to train an edit network for domains that lack ground-truth program annotations. We compare our proposed paradigm, that jointly finetunes a *one-shot* model and an edit network, against the alternative of using only a *one-shot* model, and find that our approach infers more accurate program reconstructions. Further, we find this performance gap is more pronounced when more time is spent on program search or when less training data is available. Finally, we justified the design of our method with an ablation experiment.

While our proposed approach advances the field of visual program induction, it does come with a few limitations. Compared with prior work, we need to train another network, this impacts the time required for both pretraining and finetuning stages. Moreover, the full benefit of using an edit network is best realized with a more complex program search, and as such we use search-time budgets that are slightly more costly compared with prior work. Though our formulation would offer improved performance for work-flows that can afford to spent more time on program search, it would be useful to consider potential speed-ups of our system [7]. Finally we note that our current

formulation requires access to a domain-aware *findEdits* operation that can analytically find a set of edits that realizes a transformation from a start program to an end program. While we find that our implementation generalizes across a range of visual programming domains, in future work, it would be interesting to consider to what degree this domain-aware procedure could be replaced by more general program difference algorithms [32]. Looking ahead, we believe our framework can serve as inspiration for how to train networks that learn how to edit programs without ground-truth annotations over an even wider array of program synthesis tasks.

## Acknowledgments

We would like to thank the anonymous reviewers for their helpful suggestions. Renderings of 3D shapes were produced using the Blender Cycles renderer. This work was funded in parts by NSF award #1941808 and a Brown University Presidential Fellowship. Daniel Ritchie is an advisor to Geopipe and owns equity in the company. Geopipe is a start-up that is developing 3D technology to build immersive virtual copies of the real world with applications in various fields, including games and architecture.

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

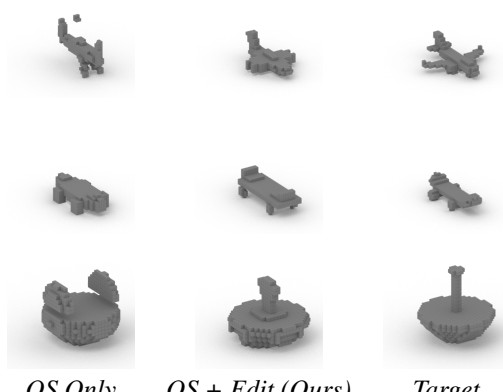

Table 3: We evaluate reconstruction accuracy for "challenge" tasks that come from concepts or categories not present in the target training set. For both layout and 3D CSG, we observe that our joint paradigm that integrates an edit network with *one-shot* models outperforms the alternative of using only *one-shot* models.

|  | Layout *cIoU* ⇑ | 3D CSG *IoU* ⇑ |
|---|---|---|
| *OS Only* | 75.8 | 60.8 |
| *OS + Edit* | **87.6** | **70.9** |

OS Only     OS + Edit (Ours)     Target

Figure 6: Qualitative reconstructions of "challenge" tasks for 3D CSG.

# A    Appendix Overview

We overview the contents of our appendices. In section B we include more experimental results. We then provide additional details on our visual programming domains (Section C), on our experimental design (Section D), on our editing operations (Section E), and on our program corruption experiments (Section F).

# B    Additional Results

## B.1    Performance on more challenging tasks

Our formulation employs a self-supervised finetuning scheme that specializes our inference networks towards a target dataset of interest. But how do our networks fare on visual inputs that are outside of these distributions? For instance, one might hypothesize that the performance gap between our joint paradigm and the *one-shot* paradigm might shrink when these approaches are given more challenging problems (e.g. when there is a large distribution gap between training and testing data).

Note though, that as we focus on local edits, our edit networks learn how to solve a local problem: given a current program and some visual target, we task our network with making any edit that would make the current program more similar to the target. Our hypothesis is that this framing should actually scale better than the *one-shot* networks when the target scenes become more complex or when they are further out-of-distribution from the training data.

Our intuition here, is that as the task complexity increases, it becomes more likely that the *one-shot* network will make mistakes. The edit network is able to account for the mistakes of the *one-shot* network and suggest local fixes that make improvements in a goal-directed fashion. When the target is out-of-distribution, even if the edit network has not seen a similar example, it can still compare the current program's execution against the target scene. Reasoning over the differences between the two states admits a more local task (as evidenced by our data efficient learning), and this property can aid in generalization.

To validate the above hypothesis, we set up an experiment to compare how our formulation (which uses a *one-shot* and edit network jointly) performs against using only the *one-shot* network for more challenging tasks in the Layout and 3D CSG domains. For the Layout domain, we evaluate the methods on scenes of new "challenge" concepts (e.g. butterflies / snowmen) that were not seen in the training / validation sets. For 3D CSG, we evaluate the methods on "challenge" shapes from other categories of ShapeNet (airplanes, knives, lamps, laptops, motorbikes, mugs, pistols, skateboards, rifles, vessels) that were not part of the original finetuning training set (chairs, tables, benches, couches).

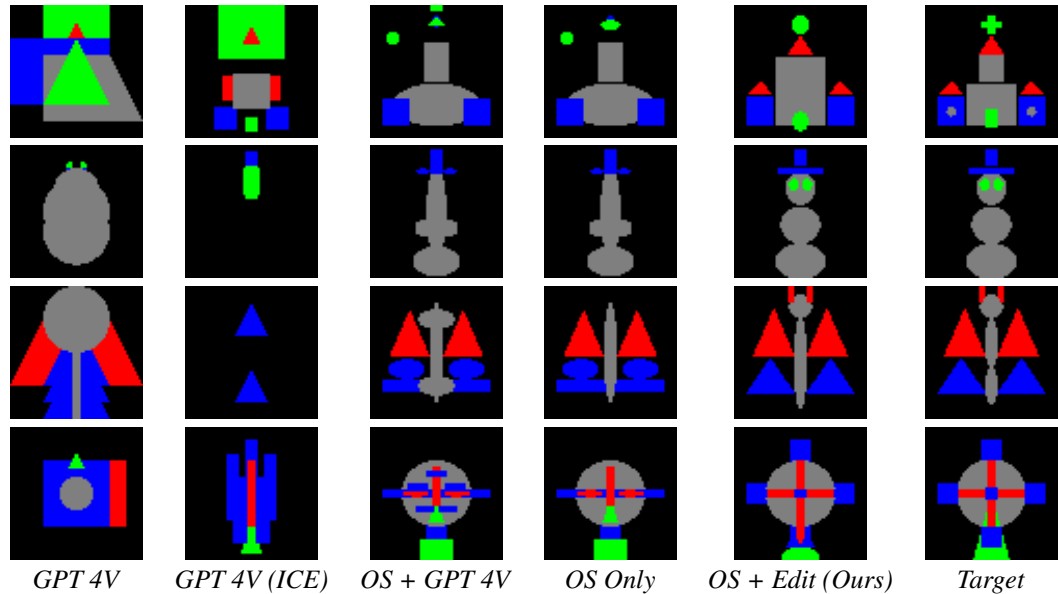

| GPT 4V | GPT 4V (ICE) | OS + GPT 4V | OS Only | OS + Edit (Ours) | Target |

Figure 7: Qualitative reconstructions of "challenge" tasks for the layout domain. We compare against GPT-4V in a zero-shot setting (column 1), when an in-content example (ICE) is provided in the prompt (column 2), and when the *one-shot* model's predicted program is provided as input (column 3). Our approach (column 5) finds more accurate reconstructions of these out-of-distribution targets (column 6) compared with using only the *one-shot* network (column 4).

Using the same models from Section 4.2, we compare the reconstruction performance for these challenge tasks. In Table 3, we report the reconstruction performance over 192 challenge tasks for the layout domain and 100 challenge tasks for the 3D CSG domain. As seen from both the quantitative and qualitative comparisons (Figures 6 and 7), it's clear that our approach, which utilizes both the *one-shot* and edit networks, outperforms using only the *one-shot* network for these more challenging program induction tasks, even when they are further *outside* the training distribution.

## B.2  Comparison to large vision-language models

As discussed in Section 2, there has been much recent research that has investigated how LLMs can aid in program synthesis tasks. Relatedly, some works have even begun to examine to what extent large vision-language models are able to understand programs that capture visual data [28]. Following similar ideas, we ran an experiment to explore how well large vision-language models (e.g. GPT-4v) are able to perform on our visual program induction tasks.

We provide some qualitative results of using GPT-4v to predict visual programs on examples from our layout domain in Figure 7. These predictions were made with a relatively straightforward prompt containing: a task-description, a description of the DSL, and the input image that should be reconstructed (zero-shot, col 1). We then tried improving this prompt by adding an in-context example of a (program, image) pair (*one-shot*, col 2). We also experimented with providing GPT-4v with a program predicted from the *one-shot* network, along with this program's execution, and asking it to edit the program to make it more similar to the target image (col 3). Please find full prompts for these experiments at https://github.com/rkjones4/VPI-Edit .

As can be seen, GPT-4v in this setting proved inferior to our proposed method (col 5). While we do not include these results to say that these sorts of large vision-language models will not *ever* be of use for this task, we do believe that these results showcase that this task is not *easily* solved with currently available frontier models.

Table 4: Ablation study on our method for the *2D CSG* domain.

| Method | Chamfer Distance ⇓ |
|---|---|
| *Ours (default)* | **0.111** |
| *No FT* | 0.321 |
| *No one-shot FT* | 0.230 |
| *No edit FT* | 0.123 |
| *No edit PT* | 0.145 |

### B.3 Method Ablations on 2D CSG domain

In Section 4.5 we presented results for an ablation experiment on the layout domain. We include additional ablation results on the 2D CSG domain in Table 4. Note that while some ablation conditions do come close to our default performance (e.g. *no edit FT*) these ablation conditions are also made possible by our contributions, as they all use an edit network. When comparing our method against an alternative without an edit network (*OS Only*, Table 1) we have consistently seen that our method offers a meaningful improvement. Below we offer some additional commentary on these results.

**No edit FT** In this ablation condition the edit network is pretrained (with synthetic random data), but is then kept frozen during the joint finetuning. As the task of the edit network is mostly local, we find that the edit network is able to achieve impressive performance even when it does not get to finetune towards data in the target distribution. That said, the edit network is still very important in this ablation condition (if it's removed then this condition becomes *OS Only*). Even though the edit network remains fixed during finetuning, it still helps to find better solutions during inner-loop inference (Alg 1, line 5), and this better training data leads to a better *one-shot* network. However, once again, the performance of the system is maximized when the edit network is also allowed to update during finetuning.

**No one-shot FT** This condition does impressively well for the layout domain. This is because even though the *one-shot* network is much worse in this setting, the edit network can overcome almost all of its mistakes, as layout is a relatively easier domain. Consider that for the layout domain, the default approach has a starting cIoU of 0.925 (initialized from the *one-shot* network, which is finetuned) which gets improved to 0.980 through improvements made by the edit network. However, the *one-shot* network of this ablation condition drops the starting cIoU to 0.88 (when it is kept frozen), and yet the edit network is still able to raise this performance all the way to 0.972 (explaining the strong reconstruction score of this condition). That said, when considering the 2D CSG ablation results in Table 4, we see that for more complex domains it is critical to also finetune the *one-shot* network, as this ablation condition achieves only a Chamfer distance of 0.230 compared with the Chamfer distance of 0.111 achieved by our default approach.

## C  Domain Details

In this section we detail the domain-specific language used for each visual programming domain.

**Layout DSL**    The layout domain creates scenes by placing colored primitives on a 2D canvas, optionally transforming them, and finally combines them together.

$$START \rightarrow UBlock;$$
$$UBlock \rightarrow \texttt{UNION}(ShBlock, UBlock) \mid ShBlock;$$
$$ShBlock \rightarrow (SymBlock \mid CBlock \mid MBlock \mid ScBlock); (PBlock \mid UBlock)$$
$$SymBlock \rightarrow \texttt{SymReflect}(axis) \mid \texttt{SymRotate}(n) \mid \texttt{SymTranslate}(n, x, y)$$
$$CBlock \rightarrow \texttt{Color}(ctype)$$
$$MBlock \rightarrow \texttt{Move}(x, y)$$
$$ScBlock \rightarrow \texttt{Scale}(w, h)$$
$$PBlock \rightarrow \texttt{Prim}(ptype)$$
$$axis \rightarrow X \mid Y$$
$$ctype \rightarrow red \mid green \mid blue$$
$$ptype \rightarrow square \mid circle \mid triangle$$
$$n \in (1, 6)$$
$$x, y, w, h \in [-1, 1]$$

In this domain, union is the only combinator operation that combines 'shape'-typed inputs by layering them on top of one another. SymReflect, SymRotate, SymTranslate, Color, Move, Scale are all transformation operations that consume a single 'shape'-typed input and apply some geometric logic to it. Prim is a special command that produces a 'shape'-typed output from only a parameter-type argument.

**2D CSG DSL**    Our 2D constructive solid geometry domain assembles complex shapes using boolean set operations. Following recent work [44] we find it useful to split each program into a set of positive sub-expressions (POS) and negative sub-expressions (NEG). Each sub-expression is allowed to take an arbitrary CSG expression, and then to form the final output all of the positive sub expressions are first unioned together, all of the negative sub expressions are then unioned together, and this second group is differenced out from the first group. This process well-matches typical procedural modeling workflows.

$$START \rightarrow POS, NEG$$
$$POS \rightarrow E, POS \mid \emptyset$$
$$NEG \rightarrow E, NEG \mid \emptyset$$
$$E \rightarrow BEE \mid TE \mid P$$
$$B \rightarrow Union \mid Difference \mid Intersection$$
$$T \rightarrow Move(F, F) \mid Scale(F, F) \mid Rotate(F) \mid Reflect(axis)$$
$$P \rightarrow \texttt{Prim}(ptype)$$
$$ptype \rightarrow square \mid circle \mid triangle$$
$$axis \rightarrow X \mid Y$$
$$F \rightarrow [-1, 1]$$

In this domain, there are three combinator operations that combine multiple 'shape'-typed inputs: union, difference and intersection. Move, scale, rotate and reflect are all transformation functions that consume a single 'shape'-typed input and apply a geometric modification. Once again, Prim is a special command that produces a 'shape'-typed argument from only a parameter-type argument.

**3D CSG DSL**   Our 3D constructive solid geometry domain generalizes the above 2D DSL.

$$START \rightarrow POS, NEG$$
$$POS \rightarrow E, POS \mid \emptyset$$
$$NEG \rightarrow E, NEG \mid \emptyset$$
$$E \rightarrow BEE \mid TE \mid P$$
$$B \rightarrow Union \mid Difference \mid Intersection$$
$$T \rightarrow Move(F, F, F) \mid Scale(F, F, F) \mid Rotate(F, F, F) \mid Reflect(axis)$$
$$P \rightarrow \texttt{Prim}(ptype)$$
$$ptype \rightarrow cuboid \mid sphere \mid cylinder$$
$$axis \rightarrow X \mid Y \mid Z$$
$$F \rightarrow [-1, 1]$$

The split between combinator, transformation and primitive creating functions is the same as in 2D CSG.

**Sampling** $L$   As previously discussed, we follow prior work and use a synthetic pretraining phase [12, 19, 34, 38]. In this pretraining phase we randomly sample programs from the above grammars. We employ simple rejection criteria to ensure these random samples are useful (e.g. no execution errors, outputs remain within the canvas, etc.), and find it effective to build in some of this rejection logic during the sampling phase (to improve the speed at which we can sample programs). All of the models we evaluate in our experiments train with the same sampling logic.

## D   Experimental Design Details

**Network details**   For our 2D domains (2D CSG and Layout) we use a 2D CNN. The image size of both domains is 64x64, but in 2D CSG there is only one input feature (occupancy) while in Layout there are three channels (RGB). The network we utilize consists of four layers, each containing convolution, ReLU, max-pooling, and dropout operations. Each convolution layer employs a kernel size of 3, a stride of 1, and padding of 1, with channel dimensions of 32, 64, 128, and 256 respectively. The CNN's output is a (4x4x256) dimensional vector, which we reshape into a (16x256) vector. This vector is then processed through a 3-layer MLP with ReLU and dropout, resulting in a final (16x256) vector that serves as a 16-token encoding of the visual input. For our 3D CNN model, we adopt a similar convolutional approach by extending all 2D convolutions to 3D. We adjust the kernel size to 4, use padding of size 2. When processing voxel grids of size $32^3$, this produces outputs of size (2x2x2x256). We pass these outputs through a 3-layer MLP to generate eight 256-dimensional visual tokens.

Our transformer networks are standard decoder-only variants. We use learned positional encodings and a hidden-dimension size of 256 and dropout of 0.1. We use networks with 8 layers and 16 heads. We set the maximum program sequence length *SL* to 128, 164, 256 for the Layout, 2D CSG, and 3D CSG domains respectively. We set the maximum edit sequence length *EL* to 32, 32, 48 for the Layout, 2D CSG, and 3D CSG domains respectively. Each prediction head (edit type, location, parameters) is modeled with a three-layer MLP with a dropout of 0.1.

**Training details**   We implement all of our networks in PyTorch [27]. All of our experiments are run on NVIDIA GeForce RTX 3090 graphic cards with 24GB of VRAM and consume up to 128GB of RAM (for 3D CSG experiments). We use the Adam optimizer [21] with a learning rate of 1e-4. For $p(z|x)$ pretraining we use a batch size of 128/128/64, for $p(e|z, x)$ pretraining we use a batch size of 128/128/32, for $p(z|x)$ finetuning we use a batch size of 20/20/20, and for $p(e|z, x)$ finetuning we use a batch size of 128/128/32 for Layout / 2D CSG / 3D CSG domains respectively. We pretrain on synthetic programs until convergence with respect to a validation set of synthetic program, for 34 / 17 / 18 million iterations, which takes 6 / 7 / 7 days for $p(z|x)$ and 70 / 30 / 25 million iterations, which takes 7 / 8 / 8 days for $p(e|z, x)$ for the Layout, 2D CSG, and 3D CSG domains respectively. We finetune each method for a maximum of 6 days or until convergence, which took 40 / 40 / 30

bootstrap rounds for the Layout, 2D CSG and 3D CSG domains. For each finetuning run we use a $P^G$ set of size 10000.

**Inference Procedure**   For our test-time inference program search we use the following population size / number of round parameters for each domain: Layout (32, 32), 2D CSG (32, 32), 3D CSG (80, 25). When using the *Os Only* method, we keep the same population / mutation general logic, but each mutation is just a randomly sampled program from $p(z|x)$. In both cases, the best reconstructing program ever seen in any round's population is returned as the 'chosen' program. The settings for this method are: Layout (32, 10), 2D CSG (32, 10), 3D CSG (25, 25). We set these parameters so that the time spent on inference per shape is even between the two modes (5, 10, 60 seconds for the three domains). For our inner-loop inference step that populates $P^{\text{BEST}}$, we use a less expensive search time budget for both modes, approximately taking (2, 5, 10 seconds for each domain respectively). We sample programs from $p(z|x)$ with top-p (.9) nucleus sampling. We sample edits from $p(e|z, x)$ with a beam search of size 3. Interestingly, we found that this sampling strategy for *Os Only* outperformed a beam-search with a beam size set to the maximum number of tokens in each $L$.

# E   Visual Program Edits

## E.1   Local Edit Operations

As described in Section 3, our network predicts local edit operations. We find it useful to constrain the set of possible edit operations as described in Section 4.5.

In order to use these local edit operations, we require a few properties of the underlying DSL. We require that it is a functional language, where each valid function has a 'shape' return type. Through a slight abuse-of-notation, we refer to functions that implicitly consume a single 'shape'-typed argument as transformation functions (e.g. *Move*), and we refer to functions that consume multiple 'shape'-typed arguments as combinator functions (e.g. *Union*). Note that as described in Section C, there may also be special functions that instantiate 'shape'-types from only non-'shape'-typed arguments (e.g. *Prim* functions).

Specifically, our formulation allows the network to predict one of the following edit operations:

- **Modify parameters** (*MP*): modifies the parameter values of a transform function. Note that this does not modify the function type (unlike *MT*). Requires additional parameter predictions to set the new values.
- **Modify transform** (*MT*): modifies a transformation function, by removing the transform and adding in a new transform with new parameters. Requires additional parameter predictions to set the new function and parameter values.
- **Add transform** (*AT*): adds a transform operator that is applied to the chosen location. Requires additional parameter predictions to specify the new function to be added and its parameters.
- **Remove transform** (*RT*): removes a transform operator and its parameter from the program. Does not require additional parameters
- **Modify Combinator** (*MC*): modifies a combinator function (e.g. changing difference to an intersection). Requires additional parameter predictions to set the new function.
- **Remove Combinator** (*RC*): removes a combinator operator (e.g. union) by specifying one input branch of the function to be completed deleted (to all of this sub-expressions leaf nodes).
- **Add Combinator** (*AC*): adds a combinator operator under the chosen transformation. Adding a combinator (such as union) requires a sequence of additional predictions to fill in one of the 'shape'-typed branches of this operator that was not previously in the program.

We once again note that each of these edit operations has a local effect. For instance, as depicted in Figure 1 adding a new transform function inserts a transform node into an already existing tree of functions. Similarly, removing a transform functions simply results in forming a skip connection from the chosen operator's parent function to the chosen operator's child function. Somewhat more arbitrary changes can be enacted by removing or adding combinators, in order to produce or remove

entire expression trees, though these are inserted or removed from specific locations. While this framing does focus on local edits, and as such our edit network makes local changes in program space, some of these changes can have dramatic effects in the execution space. For instance, consider changing a boolean operation type in CSG from difference to union.

## E.2   findEdits Algorithm

Given a starting program and an end program we develop an algorithm that analytically finds a set of edit operations that would transform the starting program into the end program. This algorithm is used to source data for the edit network, as we describe in the next section.

We design our findEdits algorithm to try to find the "minimal cost" set of edit operations that would transform a start program to an end program. Our instantiation of the algorithm works over multiple visual programming domains for the set of edit operations we consider. However, there are many alternative ways this algorithm could be instantiated, and such alterations could prove useful in helping our method adapt for very different domains. As one extreme point, consider that for general purpose programming languages, a simple "git-diff" command could be used to turn a (start, end) program pair into a set of local insert/keep/delete edits.

Our implementation evaluates valid transformations in terms of program semantics (e.g. executions) not just syntax (e.g. token matching), as there are many distinct programs in our domains that will produce equivalent execution outputs (e.g. note that the argument ordering of union for CSG languages does not change the execution output). We hypothesize that using a findEdit algorithm alternative that does not consider such "semantic-equivalences" would result in a "worse" edit network (as the patterns in the training data would be less consistent), but it would be interesting to explore how different algorithms would effect system performance in future work.

There are two main steps to this algorithm. First considering two sub-expressions $a$ and $b$, we need to find an approximately minimal set of edit operations such that applying these edit operations to $a$ would recreate the visual output of $b$. With this logic in hand, we can consider two entire programs $A$ and $B$, split them into a set of sub-expressions, $A = \{a_0, ..., a_k\}$ and $B = \{b_0, ..., b_m\}$, and then solve a matching problem to see how we should match each $a_i$ to each $b_j$ while accounting for domain-specific ordering requirements.

**Finding edits for sub-expressions**   Given two sub-expression $a$ and $b$ from one of our DSLs, we find a set of edit operations to convert $a$ to $b$ with the following recursive logic. If $a$ and $b$ have no combinator operators or order-dependant transformation functions (e.g. symmetry operations) then we can simply compare the transform functions and their arguments to see which transforms in $a$ need to be modified, added, or removed. If both $a$ and $b$ have a combinator operation, then we recurse this match on the respective sub-programs. If only $a$ has a combinator operation, we know that we need to remove one of $a$'s expression trees, so we check which of the combinator's input expression trees has the better match towards $b$. If only $b$ has a combinator operation, we know that we need to add an expression tree into $a$ with an $AC$ edit operation. The cost of this edit operation is just the length of all of the tokens of that expression tree; we evaluate the match between $a$ and each of the sub-expression within $b$ to determine which sub-expression to add with the edit operation. Any time an order dependant transform function differs between $a$ and $b$ we will either need to add, remove, or modify this transform. Note that this type of edit operation may also introduce ordering dependencies for later edit operations (which we keep track of).

**Finding a minimal matching**   From the above procedure we know the edit operations and the edit cost of transforming any sub-expression $a$ into another sub-expression $b$. We design our DSLs so that it is possible to break each program into a series of sub-expressions. For *Layout* this is done by splitting the top-level *UBlock* into the top-level *ShBlock*s. For *CSG* this is done by splitting each *POS* block into $E$ blocks and each *NEG* block into $E$ blocks. Note that there is some order dependency in this match: for CSG positive sub-expressions must be matched to other positive sub-expressions, while negative sub-expressions must be matched to other negative sub-expressions. For the *Layout* domain, *Union* is not an order invariant operator as it controls how primitives are layered on the canvas. Therefore we keep the order of *Layout* sub-expressions fixed, although we allow each sub-expression to optionally match to an empty sub-expression $\emptyset$. A match from $a_i$ to $\emptyset$ implies that $a_i$ will be removed with a *RC* edit operation, while a match from $\emptyset$ to $b_i$ implies that $b_i$

will be added with a *AC* edit operation. We consider all valid possible ways to enact this matching by calculating the cost of each sub-expression match and then extracting out a solution with the Hungarian matching algorithm [22].

### E.3   Converting edits operations to training data

From the above logic we find a set of edit operations $ES$ given input programs $A$ and $B$. As mentioned, while there may be some ordering dependencies in this set that we keep track of (e.g. adding a transform on top of newly added combinator function) this set of edit operations can be otherwise ordered arbitrarily. While many formulations are possible here we choose to convert this set into paired data for our edit network with the following procedure.

Say $ES$ contains $n$ independent edits. For each $i$ starting at 0 and ending at $n-1$ we first consider all possible ways that we could have chosen $i$ edits from $ES$. To avoid exponential blow-up, we sub-sample from this set, and choose 5 previous edit sets for each $i$. Then for each set of previous edits $pe_i$, for each next edit $e \in ES$ and $e \notin pe_i$, we add the following triplet to the training data for our edit network: the input program is $pe_i(A)$, the target visual target is $E(B)$, and the target edit operation is $e$.

### E.4   Generality of our framing

While we designed our edit operations with the task of visual program induction in mind, we believe that these operations are quite flexible. Many other functional DSLs for visual programs (and for other program synthesis tasks) could likely be subsumed directly under our framework, as long as these languages meet the criteria described in Section E.1. For instance, this set of edit operations should be able to handle any DSL expressible as a Context Free Grammar.

Under these assumptions, the edit operations we use are quite basic and make limited domain assumptions. For an input functional program, edits to transform functions allow for local edits (delete/insert/modify) that don't affect the branching factor, while edits to combinator functions allow for local edits (delete/insert) that do affect the branching factor. We employ this formulation for a few reasons: (1) it is general enough to support any program-program transformation (under our assumption set) and (2) applying any of these local edits creates a syntactically complete program that can be immediately executed.

That said, our framework and core contributions are not tied to this specific set of edit operations. Our edit network and proposed training scheme could be easily integrated with any set of local edit operations (assuming an analogous findEdits algorithm can be designed for this new set of edits). So while we believe that the set of edit operations we introduce is quite general (as evidenced by their usefulness across multiple difficult visual programming domains), we are also excited to see how our general learning-to-edit framework could be extended to even more complex DSLs and edit operations.

## F   Program Corruption

As we mention in Section 4.5 there are some high-level connections between the formulation we propose and discrete diffusion models: both do iterative error-correction and learn in a self-supervised manner to 'fix' incorrect targets. To this end, we explored alternative formulations that 'corrupted' programs. As we wanted to maintain the property that each intermediate step of the 'corruption' process is a valid program (e.g. it would not cause an executor error) we designed a domain-specific corruption process for our Layout domain. Unlike unconditional generative diffusion models that need to have strict requirements about the distribution they noise towards, we did not find this necessary in our setting as our iterative error-correcting framing is explicitly goal-directed in the form of a visual target. Specifically, our corruption process starts with an 'end' program and randomly samples 'inverse' edit operations for a random number of corruption steps. We then replace our *findEdits* step in Algorithm 1 with this corruption logic, where the start program is ignored.

While this variant is not as a performant as our default version, it still sources useful training data for our edit network. Our view is that, when possible, it is better to source these edit operations by considering start program and end program pairs, but for domains where such edit difference scripts

are hard to analytically find, this corruption variant offers an alternative. While its possible that better corruption processes could close this gap, designing them is non-trivial. From one perspective, when we want to combine *one-shot* models and edit networks at inference time, the corruption behavior we want should noise 'end' programs towards those produced by the *one-shot* model – this is exactly the distribution we get access to with the *findEdits* approach that considers program-to-program transformations. Another benefit of this formulation, is that the distribution of edit operations we train over is naturally allowed to evolve and keeps in sync automatically with the finetuned *one-shot* model. Maintaining this property with a corruption-based procedure would likely be impractical.

