# OpenReview forum: "Learning to Edit Visual Programs with Self-Supervision"
_NeurIPS.cc/2024/Conference — NeurIPS 2024 poster_

### Official Review · Reviewer_WhD9 · 2024-07-10

**Soundness:** 3
**Presentation:** 2
**Contribution:** 2
**Rating:** 5
**Confidence:** 3

**Summary:**

This paper studied the task of visual program induction (VPI), which involves inferring programs given visual targets. Compared to existing methods using one-shot generation (iteratively predicting program tokens until completion), this work proposed an Edit Network that can predict local edits that improve visual programs toward a target. The one-shot generation model and the edit model are trained jointly, which can create better-bootstrapped data for self-supervised learning. The results on three VPI domains demonstrate that the edit network can help the model's performance.

**Strengths:**

The proposed method is technically sound, and based on the quantitative and qualitative results, the edit network can improve performance.

**Weaknesses:**

1. **Lack of baselines**: This paper only compares two models, with and without the edit network, both of which are implemented by the authors. Although the model performs better with editing, it is difficult to evaluate the effectiveness of the proposed method without the context of previous methods or some other reasonable baselines, such as supervised results, zero/few-shot prompting, fine-tuning of large multimodal models, etc. In addition, I feel the improvement from Edit Network is marginal, especially for Layout and 3DCSG.

2. **Editing Operations require manual efforts**: designing the set of editing operations requires human domain knowledge. This is actually a type of supervision as humans define the set of possible edit operations, which brings extra advantages (priors) to the proposed system. In this case, the proposed method may not generalize to other domains, as you need to adjust the Editing Operations for a different problem.

3. **Reproducitity**: Many details are needed to implement this method, such as the findEdits algorithm. Even after skimming the appendix, it is still not clear to me how this algorithm works, which may pose difficulties when reproducing the results in this paper.

**Questions:**

1. Why not conduct the ablations on all three domains?
2. Can this edit network idea be applied to more general code generation tasks?

**Limitations:**

The authors discussed the limitations in the Section 5.

---

> ### Author Rebuttal · Authors · 2024-08-07
>
> Thank you for your review and interesting suggestions. We hope to address your concerns below:
>
> # Baseline Comparisons
>
> ```This paper only compares two models, with and without the edit network, both of which are implemented by the authors.```
>
> It’s true that we compare primarily against a ‘one-shot’ method to showcase the benefits of our proposed framework and edit network. However, this “baseline” is exactly the PLAD method [19].
>
> PLAD, to our knowledge, achieves state-of-the-art performance for visual program induction problems when treating the program executor as black-box (e.g. excluding methods that use gradients to differentiably optimize program parameters). Prior experiments in the literature have shown that bootstrapping methods like PLAD outperform RL based alternatives that are capable of solving VPI tasks over multiple domains [33].
>
> Further, as we noted in Section 4.2, for the 2D CSG domain our formulation achieves SOTA performance even when compared against methods that do assume access to executor-gradients. **On the test set of 2D CSG we get a Chamfer distance (CD) of 0.111 (lower is better), whereas UCSG-Net [20] gets a CD of 0.32, SIRI [12] gets a CD of 0.26, and ROAP [25] gets a CD of 0.21**. Note that as the {DSL / architecture / objective / inference procedure} differs across these various works, it’s hard to make any absolute claims from this direct comparison, nevertheless we would like to emphasize that our method’s reconstruction performance on this task is very strong in the context of the related literature.
>
> ```... supervised results, zero/few-shot prompting, fine-tuning of large multimodal models, etc.```
>
> Unfortunately we can’t compare against *supervised* alternatives as we don’t assume access to the ground-truth programs for the target dataset.
>
> *Zero/few-shot prompting* is an interesting idea, but in early experiments we struggled to get SOTA multimodal foundational models to return even moderately sensible results. To demonstrate this challenge, we include some qualitative visual program predictions made by GPT-4V in the linked pdf.
>
> These predictions were made with a relatively straightforward prompt containing: a task-description, a description of the DSL, and the input image that should be reconstructed (zero-shot, col 1). We then tried improving this prompt by adding an in-context example of a (program, image) pair (one-shot, col 2). We also experimented with providing GPT-4V with a program predicted from the one-shot network, along with this program’s execution, and asking it to edit the program to make it more similar to the target image (col 3).
>
> As can be seen in the linked pdf, all of these baselines were easily outperformed by our proposed method. While we do not include these results to say that multimodal foundation models could not be of use for this task, we do believe that these results showcase that this task is not easily solved with currently available multimodal foundation models. In fact, in our estimation, finding a way to get these sorts of models to solve these tasks in a zero or few shot setting would likely be enough of a contribution to warrant an entirely separate research paper.
>
> # Edit Operations Prior
>
> ```designing the set of editing operations requires human domain knowledge. This is actually a type of supervision as humans define the set of possible edit operations, which brings extra advantages (priors) to the proposed system.```
>
> We thank the reviewer for bringing up this important point. However, we would like to push back on calling this prior a weakness, as we actually view it as a strength of our method.
>
> From our perspective, our framework does add a new ‘prior’ beyond that used by the past work in visual program induction: the fact that instead of authoring a complete program from scratch for program synthesis tasks, it's also possible to incrementally ‘fix’ incorrect program versions towards a target.  As we discuss in the general response, our proposed framework is quite general, and as such we view this identified “advantage” as something that future work in this space can easily leverage.
>
> We note that related methods in this space have made use of similar kinds of “human domain knowledge” to build better systems. For instance, past work for simple, but general program synthesis problems has explored related sets of edit operations (keep / insert / delete / modify) for the Karel domain [14]. Moreover, library learning methods make use of complex program refactoring operations to discover re-usable sub-components that aid in program induction tasks [11]. Therefore, we don’t see our method using this extra knowledge as being unfair, but rather think that the improvement demonstrated by our method exposes a deep insight into how this challenging problem can be made more tractable.
>
> # Misc
>
> ```improvement from Edit Network is marginal, especially for Layout and 3DCSG```
>
> On this point, we would ask the reviewer to consider from our above discussion that our “one-shot” comparison actually presents a very strong baseline – so we wouldn’t view the consistent performance improvement of our method as merely marginal. We would also ask the reviewer to consider the new reconstruction results in the linked pdf on “challenge” test-sets for Layout and 3D CSG – on these harder VPI tasks the performance gap between the two approaches becomes more pronounced.
>
> ```findEdits algorithm. Even after skimming the appendix, it is still not clear to me how this algorithm works```
>
> We plan to release a reference code implementation for the entire method, which we believe will greatly aid in reproducibility and also help to inspire future efforts that can build off our system.
>
> Please let us know if there are any specific points about the algorithm description in the appendix that we can explain better – we will endeavor to make this section more clear in future versions.

---

> > ### Comment · Reviewer_WhD9 · 2024-08-10
> >
> > Thanks for the response! It addressed most of my concerns. Please update future versions with those comments.

---

> > > ### Author Response · Authors · 2024-08-12
> > >
> > > Thank you so much for engaging with our rebuttal! We're happy we were able to address your questions and we will make sure to update the paper with your comments

---

### Official Review · Reviewer_2AbF · 2024-07-13

**Soundness:** 3
**Presentation:** 3
**Contribution:** 2
**Rating:** 6
**Confidence:** 3

**Summary:**

This paper focuses on a problem in visual program induction. It proposes a new method for visual program editing, which trains an edit network along with a one-shot network with boostrapped finetuning.

**Strengths:**

-	The authors motivate and define the problem well.
-	The paper is well-written with clear illustrations, e.g. of the proposed method’s novelty (Algorithm 1) compared to the previous PLAD method.
-	The authors conducted comprehensive ablation studies on alternative formulations of their method, showing that their method achieves the best result.
-	The authors performed evaluations across three different visual programming domains.

**Weaknesses:**

-	The proposed method is somewhat but not significantly novel.
-	While the authors show performance gains of their method compared to alternatives in the ablations study, some of these gains (e.g. compared to No one shot FT and no edit FT) are surprisingly very small (<0.1). It’s not clear that all components of the proposed method are important. it would be great if the authors could provide additional analysis on why one shot FT and edit FT don't contribute much to the performance.

**Questions:**

-	Why did the authors choose to perform ablations on the Layout domain but not the other two?

**Limitations:**

-	Yes, the authors discussed a few limitations of their method, including additional training time, expensive program search, and the requirement of a domain-aware program.

---

> ### Author Rebuttal · Authors · 2024-08-07
>
> Thank you for the positive review. We are glad you found our presentation “clear” and our experiments “comprehensive”.
>
> ```The proposed method is somewhat but not significantly novel.```
>
> While novelty is hard to quantify, we do believe that our submission makes a significant impact that would be of interest to the research community.
>
> To recap, we make the following contributions:
>
> - We propose the first model for visual program induction that learns how to locally edit programs in goal-directed fashion. As mentioned in the main response section, we are hopeful that this general framing can even be useful for an even wider-array of program synthesis tasks.
>
> - We design a learning paradigm that allows us to train our edit network without access to any ground truth program supervision. We do this with a “graceful” (reBN) integration of our edit network with prior visual program induction approaches that use bootstrapped finetuning.
>
> - We introduce a set of local edit operations and a findEdits algorithm that prove useful over multiple visual programming domains (and could scale to other related DSLs).
>
> - We experimentally validate that our edit network when combined with a one-shot network offers improved reconstruction performance over using only a one-shot network under equal inference-time resources. Further, we find that the gap between these two alternatives grows as more time is spent on search, less data is available, or input tasks become more challenging (see the experimental results in our general response).
>
> So, while it is true that our method integrates nicely with and builds upon prior approaches in this area of research, we still believe that the insights we discover through this exploration constitute sufficient novelty to interest the community. As we mention in our response to WhD9, we are planning to release code and data to recreate all of our experiments, and we are excited to see how our framework could be extended to help solve related problems in the program synthesis space.
>
> ```some of these gains (e.g. compared to No one shot FT and no edit FT) are surprisingly very small (<0.1). It’s not clear that all components of the proposed method are important. it would be great if the authors could provide additional analysis on why one shot FT and edit FT don't contribute much to the performance.```
>
> Thank you for raising this question. As we mentioned in our general response, we have included more ablation condition results for the 2D CSG domain (see the linked pdf, Figure 1).
>
> In all of the ablation conditions we have experimented with, we’ve always observed the best performance with our default formulation: jointly fine-tuning both a one-shot and edit network (e.g. Ours in Table 2 of the main paper).
>
> Before further discussion, we would like to note that while some ablation conditions do come close to our default performance (e.g. “no edit FT”) these ablation conditions are also made possible by our contributions, as they all use an edit network. When comparing our method against an alternative without an edit network (OS Only, Table 1, main paper) we have consistently seen that our method offers a meaningful improvement.
>
> **no edit FT**: in this ablation condition the edit network is pretrained (with synthetic random data), but is then kept frozen during the joint finetuning. As the task of the edit network is mostly local, we find that the edit network is able to achieve impressive performance even when it does not get to finetune towards data in the target distribution. That said, the edit network is still very important in this ablation condition (if it’s removed then this condition becomes OS Only). Even though the edit network remains fixed during finetuning, it still helps to find better inferences during inner-loop inference (Alg 1, L5), and this better training data leads to a better one-shot network. However, once again, the performance of the system is maximized when the edit network is also allowed to update during finetuning.
>
> **no one-shot FT**: the “no one-shot FT” condition does impressively well for the layout domain. This is because even though the one-shot network is much worse in this setting, the edit network can overcome almost all of its mistakes because layout is one of the more simple domains. Consider that for the layout domain, the default approach has a starting cIoU of 0.925 (initialized from the one-shot network, which is finetuned) which gets improved to 0.980 through improvements made by the edit network. However, the one-shot network of this ablation condition drops the starting to cIoU to 0.88 (when it is kept frozen), and yet the edit network is still able to raise this performance all the way to 0.972 (explaining the strong reconstruction score of this condition).
>
> However, when considering the new 2D CSG ablation results, we see that for more complex domains it is critical to also finetune the one-shot network, as this ablation condition achieves only a Chamfer distance of 0.230 compared with the Chamfer distance of 0.111 achieved by our default approach.

---

> > ### Comment · Reviewer_2AbF · 2024-08-14
> >
> > Thanks for the detailed response! My questions have been addressed and I have no other concerns. I will keep my rating as it is.

---

### Official Review · Reviewer_reBN · 2024-07-13

**Soundness:** 3
**Presentation:** 4
**Contribution:** 3
**Rating:** 6
**Confidence:** 3

**Summary:**

The paper proposes a learning framework for editing visual programs. The learned edit network can be combined with the classical one-shot network, achieving better visual program induction performance under the same compute budget.

**Strengths:**

* The paper is well-motivated, with the insight that editing operations are typically local and learning such network is data efficient.
* The framework gracefully integrates with one-shot learning frameworks based on self boostrapping proposed in prior literature.
* The paper shows results on various visual domains.

**Weaknesses:**

* The visual domains in the experiments are still restricted, typically limited to simple combinations of primitives (e.g. 2D house) or a single category (e.g. chair). How to scale up the method to more complex data? For example, other categories from ShapeNet?
* The fact that editing operations are local makes the framework data efficient, but it also means that after the predicted editing operations are carried out, the program can only be improved by a small margin. This poses challenges to applying the framework to more complex domains or to scenarios where curated synthetic data have a large distribution gap compared to the target visual inputs. How to address these challenges?
* The framework requires a heuristics (findEdits Algorithm) to curate data. This seems reasonable, but as it can have a large impact on performance, more explanations on why designing the algorithm this way would be helpful.
* The findEdits Algorithm assumes that there are two types of functions: transforms and combinators. Is this a generic enough formulation? Are there cases when there are other types of functions, or even helper functions, for a DSL with richer syntax?

**Questions:**

* Why is Modify transform (MT) necessary? Why is it not equivalent to Remove transform (RT) followed by Add transform (AT)?
* How to sample start programs and end programs? Are they randomly sampled subexpressions from the DSL?

**Limitations:**

Limitations are discussed.

---

> ### Author Rebuttal · Authors · 2024-08-07
>
> Thank you for your positive review and interesting questions.
>
> ```How to scale up the method to more complex data? For example, other categories from ShapeNet```
>
> Thank you for the suggestion! Please see our general response for new experimental results on more challenging test-set tasks.
>
> ```How to sample start programs and end programs? Are they randomly sampled subexpressions from the DSL?```
>
> To get training data for our edit network we use our findEdits operation which finds a set of edit operations that transforms a start program into an end program.
>
> During edit network finetuning (Alg 1, L10), we source “end programs” by sampling our generative model of programs, p(z) (Alg 1, L6).
>
> During edit network pretraining (Alg 1, L1), we source “end programs” by randomly sampling synthetic data from the DSL (using the same sampling scheme used in one-shot network pretraining).
>
> To source a “start program” for each “end program”, we use the one-shot network p(z|x) and condition on the executed version of the “end program” (Alg 1, L8).
>
> In this way the (start, end) program pairs we use to train the edit network well-match the distribution of programs that we will task the edit network with improving at inference time (predictions from the one-shot network).
>
> ``` The framework requires a heuristics (findEdits Algorithm) to curate data. This seems reasonable, but as it can have a large impact on performance, more explanations on why designing the algorithm this way would be helpful.```
>
> Yes, this is a fair point – as from above, we need the findEdits algorithm to convert pairs of (start, end) programs into edit operations so that we can train the edit network.
>
> In the submission, we develop one instantiation of a findEdits algorithm that works over multiple visual programming domains for the set of edit operations we consider. However, there are many alternative ways this algorithm could be instantiated, and such alterations could prove useful in helping our method adapt for very different domains. As one extreme point, consider that for general purpose programming languages, a simple “git-diff” command could be used to turn a (start, end) program pair into a set of local insert/keep/delete edits.
>
> We design our findEdits algorithm to try to find the “minimal cost” set of edit operations that would transform a start program to an end program. We evaluate valid transformations in terms of program semantics (e.g. executions) not just syntax (e.g. token matching), as there are many distinct programs in our domains that will produce equivalent execution outputs (e.g. note that the argument ordering of union for CSG languages does not change the execution output). We hypothesize that using a findEdit algorithm alternative that does not consider such "semantic-equivalences" would result in a “worse” edit network (as the training data would be noisier), but we agree that it would be interesting to explore how different algorithms would effect system performance in future work.
>
> ```Why is Modify transform (MT) necessary? Why is it not equivalent to Remove transform (RT) followed by Add transform (AT)?```
>
> Good catch! The MT operation is actually not strictly necessary, because, as you say, a RT operation can be coupled with an AT operation to produce the same change.
>
> That said, a single MT operation is more efficient (from the above “minimal cost” perspective) compared with coupling an RT with an AT operation. Our hypothesis here is that it would be easier for the edit network to make this kind of change with a single edit operation, simplifying training and making inference time search more efficient.
>
> ```it also means that after the predicted editing operations are carried out, the program can only be improved by a small margin.```
>
> Beyond our discussion in the general response, we also wanted to revisit an aspect of this point.
>
> While it’s true that by focusing on local edits, our edit network will make local changes in program space, some of these changes can have dramatic effects in the execution space. For instance, consider changing a boolean operation type in CSG from difference to union. Further, while each operation is local, the AC (add combinator) operation does introduce an entirely new sub-expression, while the RC (remove combinator) operation does remove an entire sub-expression; so even some of the “local” edit operations can create fairly dramatic program space changes.
>
> On this note, in our ablation conditions we did consider other "obvious" formulations for how to predict edits, and found both of these alternatives offered worse trade-offs compared with predicting our local edits operations. Looking forward, we agree that it would be interesting to explore extensions that allow the edit network to choose between making more "local" or "drastic" edits to the input program. Our framework would likely serve as a good building block for this kind of investigation, as one could consider composing edit operations returned by the findEdits procedure into ‘target edits’ of different magnitudes. While some work would need to be done on the modeling side to figure out what magnitude of edit should be applied when, we are hopeful that our method can serve as a useful reference for future endeavors in this research area.

---

### Author Rebuttal · Authors · 2024-08-07

We thank the reviewers for their encouraging and helpful comments.

We are glad that reviewers found our proposed framework “well-motivated” (2AbF) and insightful (reBN). Reviewers remarked that our submission was “well-written” (2AbF) and “technically sound” (WhD9), and further commented that our formulation has a “graceful” (reBN) interplay with prior work on visual program induction. Overall, reviewers noted our method’s strong empirical performance over alternatives, as our formulation offers better “performance under the same compute budget” (reBN), in terms of both “quantitative and qualitative results” (WhD9) over multiple domains (reBN, 2AbF, WhD9).

# Task Complexity

We thank reBN for bringing up an interesting point, in that there may be concern about how our proposed edit network would fare on more challenging problems, where, for instance, there is a large distribution gap between training and testing data. We actually think this is exactly the sort of problem setting where learning to make local edits can offer improved performance!

As we focus on local edits, our edit networks learn how to solve a local problem: given a current program and some visual target, we task our network with making any edit that would make the current program more similar to the target. Our hypothesis is that this framing should actually scale better than the one-shot networks when the target scenes become more complex or when they become further out-of-distribution from the training data:

- As the task complexity increases, it becomes more likely that the one-shot network will make mistakes. The edit network is able to account for the mistakes of the one-shot network and suggest local fixes that make improvements in a goal-directed fashion.

- When the target is out-of-distribution, even if the edit network hasn’t seen a similar example, it can still compare the current program’s execution against the target scene. Reasoning over the differences between the two states admits a more local task (as evidenced by our data efficient learning), and this property can aid in generalization.

**Explanation of new results**

To validate the above hypothesis, we ran additional inference experiments to compare how our formulation (which uses one-shot and edit network jointly) performs against using only the one-shot network for more challenging tasks in the layout and 3D CSG domains.

For the layout domain, we evaluate the methods on scenes of new “challenge” concepts (e.g. butterflies / snowmen) that were not seen in the training / validation sets.

For 3D CSG, we take reBN’s suggestion and evaluate the methods on “challenge” shapes from other categories of ShapeNet (airplanes, knives, lamps, laptops, motorbikes, mugs, pistols, skateboards, rifles, vessels) that were not part of the original finetuning training set (chairs, tables, benches, couches).

Using the same models from the submission, we compare the reconstruction performance for these challenge tasks in the linked pdf (with the inference procedure described in the main paper). In Table 1, we report the reconstruction performance over 192 new challenge tasks for layout and 100 new challenge tasks for 3D CSG.

From both the quantitative and qualitative comparisons (Figures 2 and 3 in the linked pdf), it’s clear that our approach, which utilizes both the one-shot and edit networks, outperforms using only the one-shot network for these more challenging program induction tasks, even when they are further ‘outside’ the training distribution. We would be happy to add these results and discussion to the main paper.

# Generality of Edit Operations

Some reviewers (reBN and WhD9) had questions about how our proposed set of edit operations would generalize.

While we design our edit operations with the task of visual program induction in mind, we believe that these operations are quite flexible. Many other functional DSLs for visual programs (and for other program synthesis tasks) could likely be subsumed directly under our framework, as long as these languages meet the criteria described in Appendix D. For instance, this set of operators should be able to handle any DSL expressible as a CFG.

Under these assumptions, the edit operations we use are quite basic and make limited domain assumptions. For an input functional program, edits to transform functions allow for local edits (delete/insert/modify) that don’t affect the branching factor, while edits to combinator functions allow for local edits (once again delete/insert/modify) that do affect the branching factor.

We employ this formulation for a few reasons: (1) it is general enough to support any program-program transformation (under our assumption set) and (2) applying any of these local edits creates a syntactically complete program that can be immediately executed.

That said, our framework and core contributions are not tied to this specific set of edit operations. Our edit network and proposed training scheme could be easily integrated with any set of local edit operations (assuming an analogous *findEdits* algorithm can be designed for this new set of edits). So while we believe that the set of edit operations we introduce is quite general (as evidenced by their usefulness across multiple difficult visual programming domains), we are also excited to see how our general learning-to-edit framework could be extended to even more complex DSLs and edit operations.

# Ablation Study Domains

(2AbF, WhD9) In our submission, we ran our ablation study over the layout domain, as it was the only domain where we had a “corruption” procedure (designing such procedures are non-trivial).

As there were some questions regarding the “pretraining and finetuning” ablation results, we also report the performance of these conditions for the 2D CSG domain in our linked pdf (Figure 1). We find largely similar results as we did on the layout domain, but see our response to 2AbF for further discussion.

---

### Decision · Program_Chairs · 2024-09-25

**Decision:**

Accept (poster)

**Comment:**

The authors propose a self-supervised bootstrapping approach for visual program induction that uses an edit network in conjunction with one-shot model. The work received positive ratings from all reviewers (2WA, BA). The reviewers appreciated that the paper is technically sound, is well written and motivated, and has comprehensive ablations. Initial concerns raised by the reviewers included results on limited visual domains, necessity of all components, and limited baseline comparisons. The reviewers were generally satisfied with the authors' response to these concerns.

The AC agrees with the overall positive sentiment of the reviewers and is happy to recommend "accept". The AC encourages the authors to incorporate all discussion to revise their draft and consider extensions to natural images for future work. Congratulations to the authors for a well presented submission that explores an interesting idea!